# Detection of large extracellular silver nanoparticle rings observed during mitosis using darkfield microscopy

**Robert M. Zucker**[ID]\*, **Jayna Ortenzio, Laura L. Degn, William K. Boyes**[ID]

U.S. Environmental Protection Agency, Office of Research and Development, Center for Public Health and Environmental Assessment, Public Health and Integrated Toxicology Division, Reproductive and Developmental Toxicology Branch, Research Triangle Park, Durham, NC, United States of America

\* zucker.robert@epa.gov

**Data Availability Statement:** All relevant data are within the manuscript and its Supporting Information files.

**Funding:** The funding for this research was provided through the intermural research funds of

## Abstract

During studies on the absorption and interactions between silver nanoparticles and mammalian cells grown in vitro it was observed that large extracellular rings of silver nanoparticles were deposited on the microscope slide, many located near post-mitotic cells. Silver nanoparticles (AgNP, 80nm), coated with citrate, were incubated at concentrations of 0.3 to 30 µg/ml with a human-derived culture of retinal pigment epithelial cells (ARPE-19) and observed using darkfield and fluorescent microscopy, 24 h after treatment. Approximately cell-sized extracellular rings of deposited AgNP were observed on the slides among a field of dispersed individual AgNP. The mean diameter of 45 nanoparticles circles was 62.5 +/-12 microns. Ring structures were frequently observed near what appeared to be post-mitotic daughter cells, giving rise to the possibility that cell membrane fragments were deposited on the slide during mitosis, and those fragments selectively attracted and retained silver nanoparticles from suspension in the cell culture medium. These circular structures were observable for the following technical reasons: 1) darkfield microscope could observe single nanoparticles below 100 nm in size, 2) a large concentration ($10^8$ and $10^9$) of nanoparticles was used in these experiments 3) negatively charged nanoparticles were attracted to adhesion membrane proteins remaining on the slide from mitosis. The observation of silver nanoparticles attracted to apparent remnants of cellular mitosis could be a useful tool for the study of normal and abnormal mitosis.

## Introduction

Nanotechnology is a rapidly growing segment of commerce that is projected to grow into a global market with hundreds of billions of dollars in revenue (Lux Corporation, 2014). A key aspect of the nanotechnology revolution is the production of engineered nanoparticles (usually defined to have a particle size between 1–100 nm), which have novel properties that are exploited in creative new products and applications, but which may also pose novel concerns for both exposures and hazards [1]. Among many complex interactions of nanomaterials with biological systems is a central issue of being able to observe nanomaterials and characterize

the Office of Research and Development, U.S. Environmental Protection Agency. This research was supported in part by an appointment to the ORISE Research Participation Program for the U.S. Environmental Protection Agency, Office of Research and Development, administered by the Oak Ridge Institute for Science and Education through an interagency agreement between the U. S. Department of Energy and U.S. EPA.

**Competing interests:** The authors have declared that no competing interests exist.

their distribution inside mammalian cells. This fundamental ability is problematic because the small size of nanoparticles is below the optical resolution limits of standard light microscopes. Scanning or transmission electron microscopes may have the optical resolution to observe nanoparticles [2,3], but typically are expensive, have limited access, and their operation is time intensive. Several investigators have used fluorescently labeled nanomaterials or quantum dots to track the intracellular movement and location of nanomaterials [4–9], however not all particles are fluorescent, and the addition of fluorescent tags might alter the behavior of particles. The inability to regularly observe nanomaterials inside cells limits progress toward understanding their behavior and predicting the actions of novel untested materials. In addition to an increased understanding of the effects of nanoparticles on cells, these studies may have an ancillary benefit of improving the ability to study the physiology of the cells themselves.

Previously, flow cytometry and dark field microscopy were used to investigate nanoparticle interactions with cells [7,10–20], including $TiO_2$ and silver nanoparticle (AgNP) uptake into a human-derived retinal pigment epithelial cell line (ARPE-19) [16–18]. The incubation of AgNP with APRE-19 cells resulted in a dose-dependent increase of particles in cells observed by darkfield microscopy [13,16,17,21,22]. In darkfield microscopy, the use of a xenon light source is advantageous because xenon has a shorter wavelength spectrum than conventional illuminants, which enables better resolution of small nano-sized objects. Under darkfield imaging, $TiO_2$ or AgNP were observed to enter the cytoplasm of ARPE-19 cells, and form peri-nuclear agglomerations that increased in size as the exposure concentration or time increased [16,17,21].

The present study extends our previous nanoparticle research by demonstrating circular structures of nanoparticles on slides. These circular structures were approximately one half to two thirds of the size of normally log growing cells, and they were frequently located in the vicinity of post-mitotic daughter cells.

## Materials and methods

### Cell culture

Human-derived retinal pigment epithelial cells (ARPE-19, passage 28) (ATCC, Manassas, Virginia) were grown in T75 culture flasks in a 1:1 mixture of Dulbecco's Modified Eagle's Medium and Ham's F-12 Nutrient Mixture (DMEM/F-12) with 10% fetal bovine serum (FBS). After reaching confluence, cells were trypsinized (0.05% trypsin, EDTA 0.02%, Sigma), and plated on 4-chambered glass tissue culture slides (1 ml cell suspension per chamber, $2\times10^5$ cells/ml). Cells were incubated for 24 h (37°C, 5% $CO_2$) after plating, and were treated with nanoparticles while in a log growth phase prior to reaching confluence. After treatment, cells were maintained for a further 24 h before staining and fixation for microscopy [13,16,17].

### Silver nanoparticles

Aqueous suspensions (1 mg/ml) of 80 nm citrate coated AgNP were obtained from NanoComposix (San Diego CA). The supplier provided lot-specific characterization of particles including TEM images, particle diameter, surface area, mass concentration, endotoxin concentration, hydrodynamic diameter, pH of the solution, and chemical purity. Characterization of physical properties of silver nanoparticles by NanoComposix can be found at the following website: (https://nanocomposix.com/pages/silver-nanoparticles-physical-properties) The AgNP were diluted further in cell culture media (1:1 DMEM/F-12 with 10% FBS) to concentrations between 0.3 and 30 µg/ml. ARPE-19 cells were incubated with suspensions of AgNP for 24 h prior to observation.

## Staining, fixation, and mounting

Prior to fixation, the cells were counterstained with CellMask Orange plasma stain (C10045) to identify the cell's cytoplasmic area. The cells were washed and fixed with an equal amount of warm 4% paraformaldehyde (PF) in phosphate-buffered saline (PBS). To identify intra-cellular structures, the cells were transfected with Golgi GFP BacMan 2.0 (C10592), lysosome-RFP (C10597), mitochondria-GFP (C10600), actin-RFP (C10583), or endoplasmic reticulum-GFP BacMan (C10590) (Invitrogen, Eugene Oregon). Prolong Gold, containing 10 μg/ml DAPI (P36935), was used to mount the slides and stain the nuclei. After the mounting medium dried, slides were sealed and then observed with a combination of dark field and fluorescence microscopy.

## Microscopy

A Nikon E-800 upright microscope, a Nikon Ni upright microscope, and a Nikon Ti2 inverted microscope were used to observe cells in darkfield and fluorescence illumination. The E-800 microscope had space for five filter cubes while the Nikon Ni and Ti2 had space for 6 fluorescence filter cubes. The fluorescence excitation cubes were for DAPI, FITC, TRITC, and Cyan GFP. The fifth space in the E-800 cube holder or 6[th] space in the Nikon Ni or Ti2 was intentionally left without a cube to acquire a clean dark field image without distortion from filters. The Nikon E-800, Ni and Ti2 microscopes contained bandpass and long pass filters to excite fluorescence probes with blue (480 nm) or green (546nm) light.

The dark field image was about 100 times brighter than the fluorescence image using common exposure times. Due to the brighter image obtained from the AgNP with xenon light, there was a possibility of a slight bleed through of emitted light using any of the four filter cubes. The fluorescent and dark field images were therefore taken sequentially at different exposure times and then combined using Nikon Elements 5.1 software. Co-localization of the optical system was established with 0.5 μm Tetra spec beads and with 1 μm and 15 μm multi-wavelength fluorescent ring beads (Molecular Probes, Eugene Oregon). The xenon light supply was used with the dark field images to obtain a bright light source that was optimized for the shorter blue wavelength excitation yielding better resolution. A GG 420 or GG 435 filter was put in the eyepieces to protect the user's eyes from possible UV damage from the Xenon light source [13,16].

A 60x Plan Fluor lens with an iris diaphragm to control the numerical aperture (NA) between 0.55 and 1.25 was used for imaging. Cellular details could be observed with this magnification while the background scatter was controlled by adjusting the iris diaphragm. The lower NA yielded good depth of field for bright AgNP, and the higher NA (1.25) yielded bright fluorescence images of the cellular components. By balancing the fluorescence and dark field signals, a sequential image could be acquired for combination with the same NA setting (approximately 0.8 NA). During this study, the dark field images were obtained using the following lenses: Plan Apo 20x, (NA 0.75), 20x multi-immersion (NA 0.75), and 60x Plan Fluor with iris (NA 0.55–1.25). The NA of lenses with an oil darkfield condenser was below about 0.95 to provide proper illumination without reflection for dark field images of AgNP. Most of the microscopy images were acquired with the 60x Plan Fluor lens with an iris adjustment of 0.7 NA to optimize both the fluorescence and dark field in the same image [13,16].

## Measurements of nanoparticle circles

The microscope objectives (20x and 60x) were calibrated in microns using a Nikon micron measurement slide. The nanoparticle rings were mostly circular. Occasionally, some of the rings appeared to be more oblong than circular and therefore measurements in both the

vertical and horizonal directions were made on each circular structure. A total of 90 perpendicular length measurements were obtained from 45 relatively circular objects using Nikon Elements 5.13 software. The thickness of the ring and the size of the hole were also measured on some of the nanoparticle circles. In addition, the cellular and nuclear diameter from the longest axis were measured. The mean and standard deviation of the diameter measurements were calculated using Excel.

## Results

In these experiments the dose range varied from 0.1 μg/ml to 30 μg/ml. However, for clarity of the figures only cells treated with 3–10 μg/ml of 80 nm citrate AgNP are presented in the manuscript. A combination of fluorescent and dark field images of cells treated with 3 μg/ml 80 nm citrate AgNP is presented in Fig 1 The AgNP appear as bright white spots against a black background in the dark field images. The AgNP that settled onto the slides outside the cells appeared to be randomly distributed and were relatively faint, while those inside the cells were considerably brighter than the nanoparticles located outside the cells. Occasionally, the background of the slide contained large extracellular structures that were observable when scanning the field with a low power 20x objective (Fig 1). These round circular structures were observed more frequently on slides that were incubated with higher concentrations of nanoparticles (3–10 μg/ml). These circular structures consisted of nanoparticles deposited in ring formations. The rings appeared to be about one third smaller than the size of intact cells. The outer diameter of the circles was approximately 60 μm. Observation of the ring structures revealed that the thickness of the ring was approximately 18 μm while the diameter of the hole was about 25 μm. No circular structures were observed on slides containing AgNP without cells; similarly, no circular structures were observed on slides of cells without AgNP. This indicates the circular structures were formed due to an interaction between the cells and AgNP.

Fig 2 shows four different fields at 600x, each containing a circular structure associated with a pair of newly divided cells that were either in direct contact with the nanoparticle circle or located nearby the circle. The cells were stained for mitochondria (GFP-mitochondria, green), nuclei (DAPI, blue) and cytoplasm (CellMask orange, red). The individual particles inside the cell and comprising the ring could be observed at higher magnification (600x) using a 60x Plan Fluor objective containing an iris diaphragm (Fig 2A). The size of the particles outside the circular structure appeared to be like the size of the particles comprising the circular structures. However, the particles inside the cell were brighter than the particles outside the cell. Due to the roundness of the new post-mitotic cells observed in panels B, C and D, there is cytoplasm located on top of the nuclei, resulting in a composite fluorescent image that is a purple color representing the combination of red cytoplasm and blue nuclei. The cells in Fig 2A–2C are located on top of the nanoparticle circle. In contrast, the cells in Fig 2B–2D appear on top or inside of the circle.

Fig 3 shows a thin cell located adjacent to a circular structure of AgNP derived from 3ug/ml AgNP citrate incubation. To better visualize the cellular organelle fluorescence and AgNP, the images are presented in four panels consisting of two individual fluorescence images (Fig 3C and 3D) and one darkfield image of AgNP (Fig 3A) along with a composite of all of the images (Fig 3B). The circular structure in Fig 3 contained no visible fluorescent signals (white, Fig 3A). The cell adjacent to the circle contained fluorescence signals staining the cytoplasmic (CellMask Orange (Fig 3D), mitochondria (Fig 3C, GFP-green), and nucleus-DAPI (Fig 3B, blue).

Examination of some of the large circular AgNP structures at 60x revealed that there was a small amount of cytoplasmic staining observed with the 545 nm excitation illumination of CellMask Orange membrane stain around the circumference of some of these circular

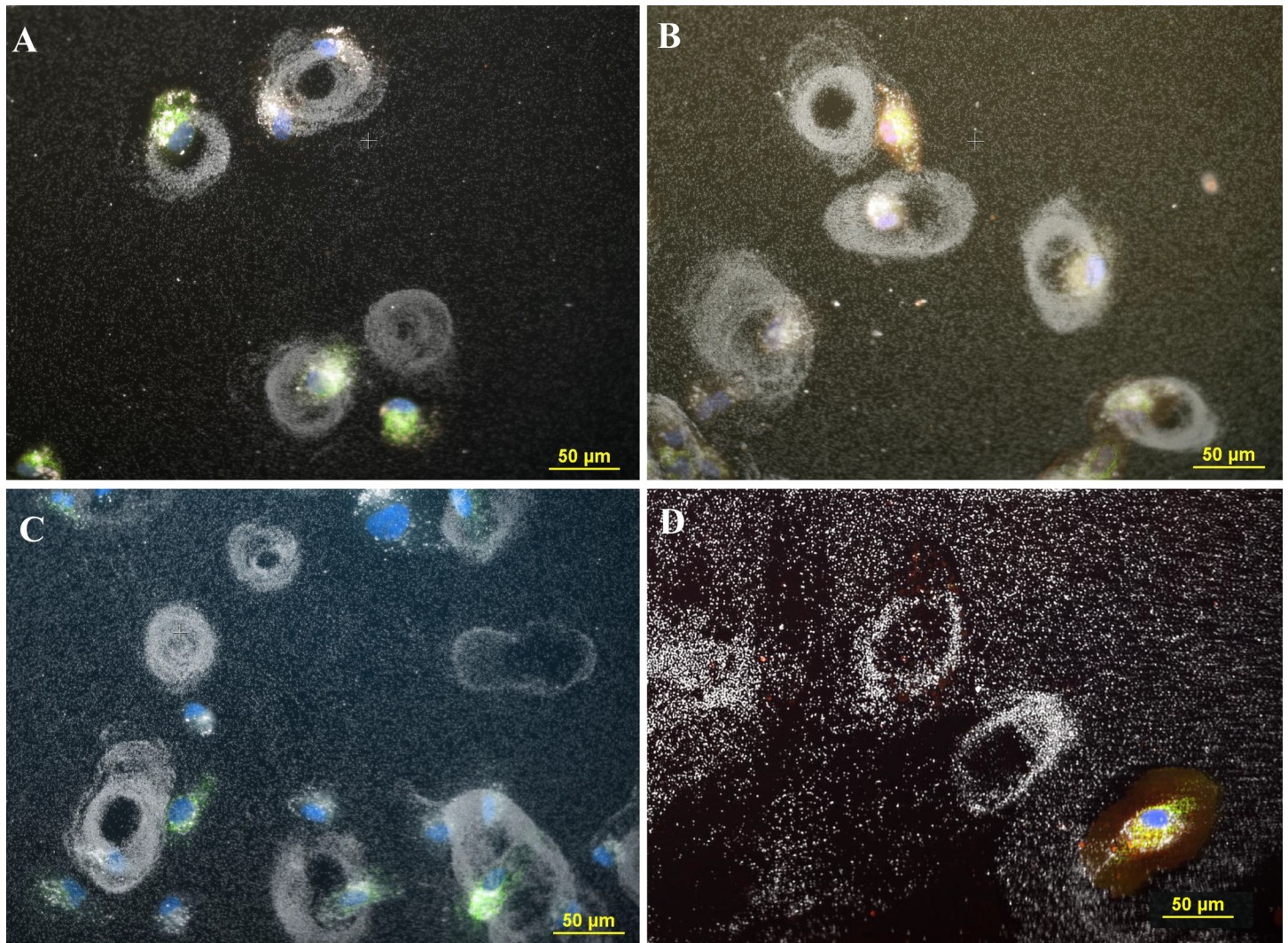

**Fig 1. The images in Fig 1 (all panels) show ARPE-19 cells incubated with 3 µg/ml AgNP citrate.** The cells were transfected with mitochondria-GFP (green), stained with CellMask Orange plasma membrane stain (C10045, red), fixed with 4% PF, and mounted with Prolong gold that contained DAPI to stain the nuclei (blue). This allowed for the visualization of cells, and nanoparticles (white) due to reflected light in the darkfield images. The images were observed using a 20x multi-immersion objective (NA 0.75). The images in each panel are a darkfield image and fluorescence images that were acquired sequentially and then combined into one composite image. Panel A. Image of four rings of silver nanoparticles on the cell culture slide. Cells are nearby or adjacent to the rings. Magnification 200x. Panel B. Image of five rings and adjacent cells. Magnification 200x. Panel C. Image of six rings and multiple cells. Magnification 200x. Panel D. Image of two circular structures consisting of nanoparticles, and a cell that was stained with CellMask Orange plasma membrane stain (C10045), transfected with mitochondria-GFP (green) and stained with DAPI (blue) to reveal the nucleus. The cell in the Panel D is about the same size as the nanoparticle circles in the image. Magnification 200x.

structures (Fig 4). The image illustrated in Fig 4B was acquired from the identical field with fluorescence excitation optics using the 545-nm wavelength light to excite the CellMask Orange plasma membrane dye and a TRITC fluorescence cube. The partial red ring structure observed in Fig 4B appears to represent membrane or cytoplasmic components that remained on the slide after the cell contracted during the mitotic process. When the two images were superimposed, the red fluorescence rim (Fig 4B) was co-localized with the white nanoparticle ring (Fig 4A). The combination of these two images (not shown) suggested that the fluorescent emission image observed in Fig 4B was related to the nanoparticles in Fig 4A. This ring pattern may have been the result of cytoplasmic/membrane remnants left on the slide during the mitotic process. Apparently, the ring structure had some molecules that were stained with the

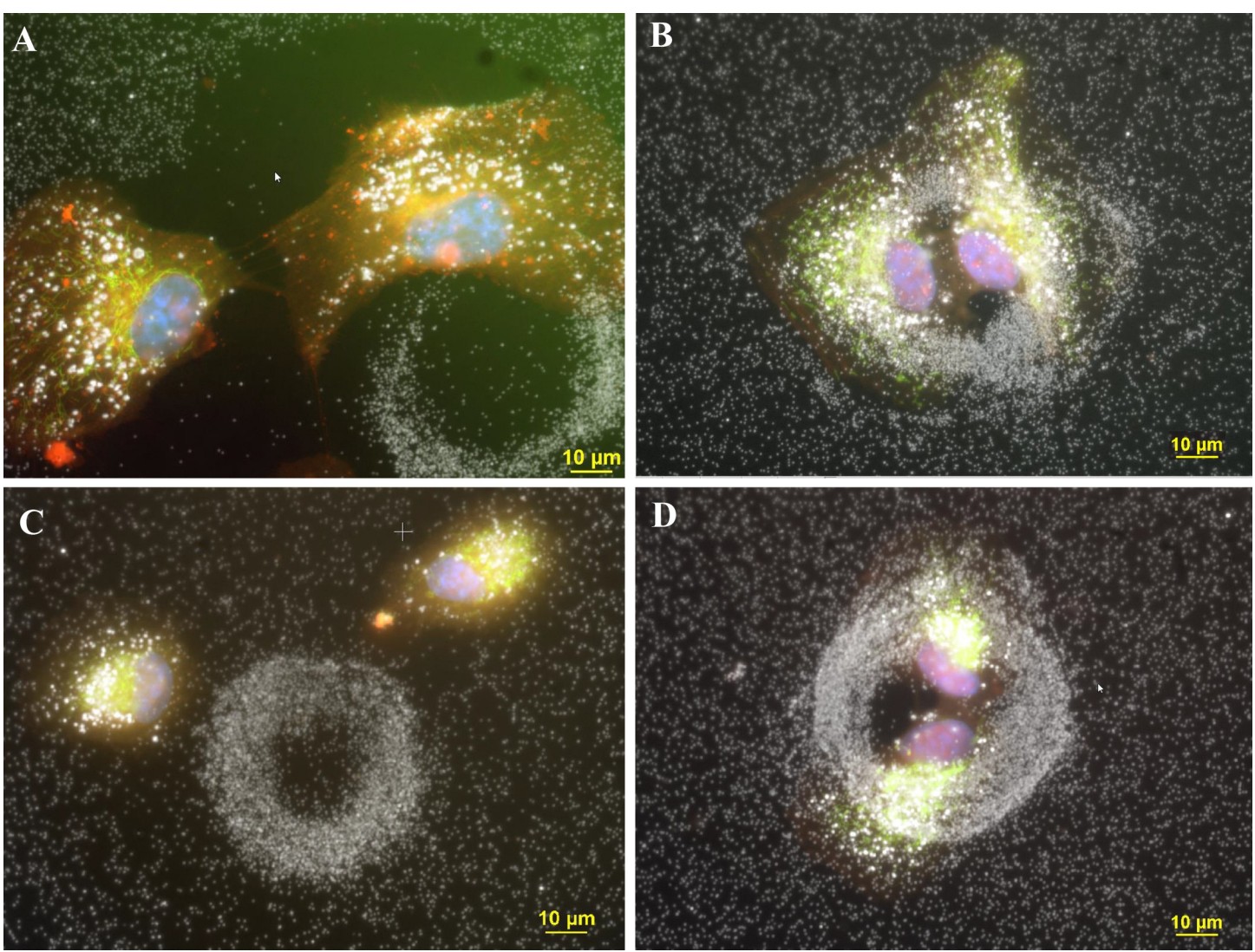

**Fig 2. Four different fields of cells treated with 3ug/ml AgNP citrate contained circles that was visualized obtained with darkfield optics using a 60-x plan Fluor objective with an iris diaphragm (NA 0.55–0.90).** Panel A. Two cells are located outside the nanoparticle circle that have an accumulation of intracellular nanoparticles that are brighter than the extracellular nanoparticles. The particles in the extracellular ring have similar brightness to other extracellular particles. Magnification 600x. Panel B. Two cells in mitosis is visible inside the nanoparticle circle. The cells were stained with CellMask Orange plasma stain (red) to reveal the cytoplasm and DAPI (blue) to reveal the nucleus. Since the mitotic cells are round, some of the colors of red cytoplasm have blended with the blue nuclei to yield a purple color. Cells were transfected with Mito-GFP (green). The two apparently mitotic cells are surrounded by a single nanoparticle ring structure. The circle diameter size is 54 µm, the center hole diameter is 30 µm and nuclear diameter is 18.7 µm. Magnification 600x. Panel C. Two post-mitotic daughter cells were observed near a large ring structure. The circle diameter is 56 µm, hole diameter is 23 µm, and circle wall thickness is 16 µm. Magnification 600x. Panel D. Two post-mitotic daughter cells were observed within a larger ring structure The circle diameter is 62 µm, the center diameter is 30 µm, and circle rim thickness is 15 µm. Magnification 600x.

CellMask Orange plasma membrane stain. These molecules may be part of membrane structures that preferentially attracted extracellular nanoparticles from the medium. Incubation with the positively charged 80 nm branched polyethyleneimine (AgNP-bPEI) did not show this circular ring pattern [25].

## Measurement of circular ring sizes

Forty-five circular AgNP rings were measured in two perpendicular directions using either a 60x or 20x objective lens. A histogram of the sizes of the circular structures is shown in Fig 5.

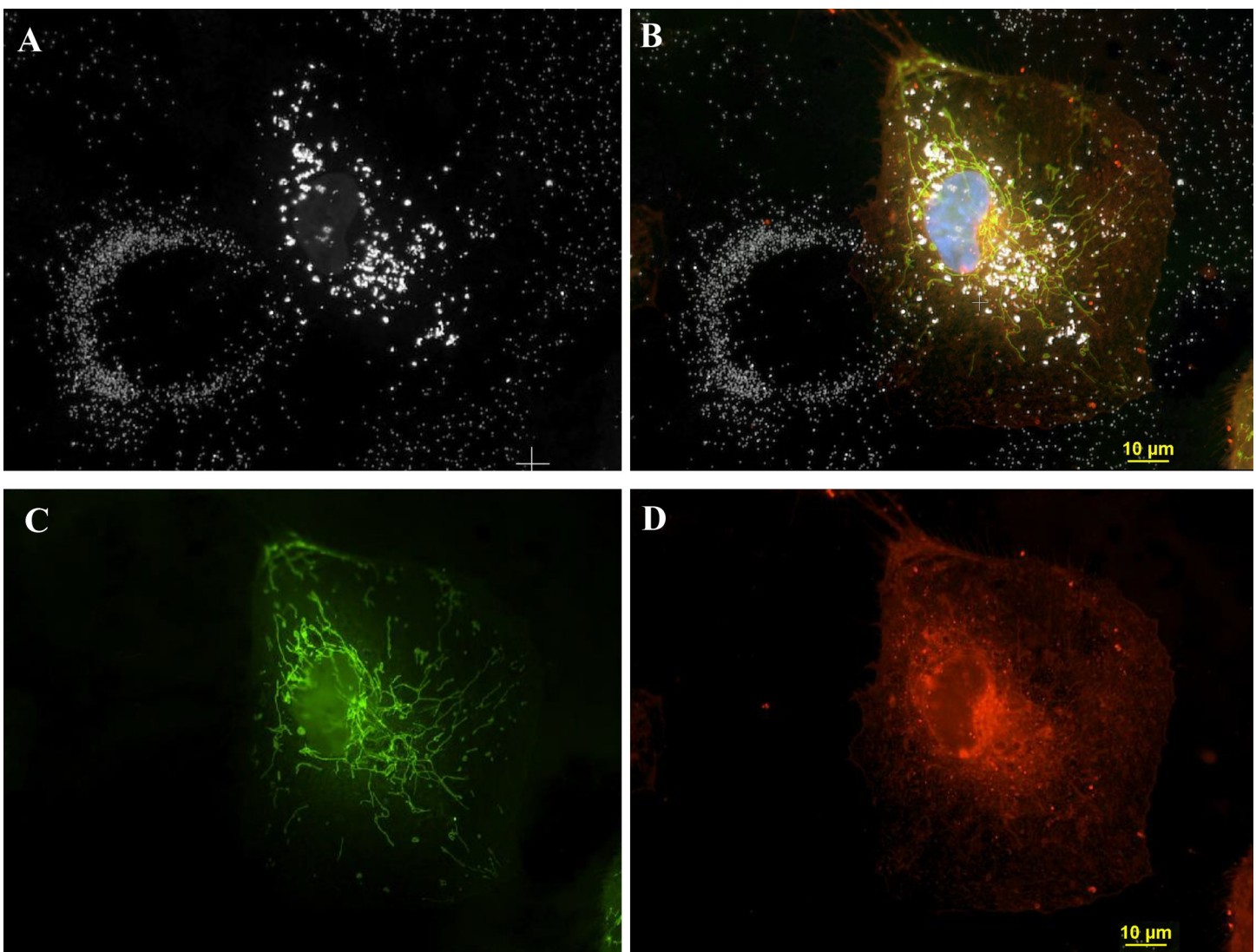

**Fig 3. The four-panel figure shows the individual dark field and fluorescent images that were combined into a composite image.** A 60-x plan Fluor objective with an iris diaphragm (NA 0.55–0.90 was used to obtain the image. Panel A. A darkfield micrograph of a cell with an adjacent ring structure. The diameter of the circle is 53 µm and the hole is 32 µm. The longest direction is 90 µm and the nuclear diameter is 24 µm. The intensity of nanoparticles contained in the cell are brighter than the ones contained in the ring structure. Panel B. A composite image consisting of 3 fluorescent images and a darkfield image The composite image was deconvolved using Nikon Autoquant 2D "blind" program, which increased the resolution of the fluorescent probes, especially the Mito- GFP. Panel C. A fluorescent image showing mitochondrial Mito-GFP. (green). Nikon Autoquant 2D "blind" program was used which increased the resolution of the Mito-GFP probe. Panel D. A fluorescent image of the cytoplasm Cell Mask Orange membrane stain. Magnification 600x.

The mean diameter of the 45 rings including 2 perpendicular measurements was 62.5 microns with a SD of 12. On the same slides, 12 nuclei, which were usually oblong, had a mean value of 18 microns with a SD of 4 microns. Measurement of the individual components of the circular structure showed a circular wall of 18 µm and a hole diameter of approximately 25 µm. The nanoparticle sizes measured under darkfield optics were about 0.37 µm outside the cell and 0.61 µm inside the cell. This increase in size suggests clumping of nanoparticles inside the cell associated with the increased amount of scattered light. The size of particles within the ring could not be determined easily as they were clumped and concentrated, but they did appear to be similar in size to the other extracellular particles.

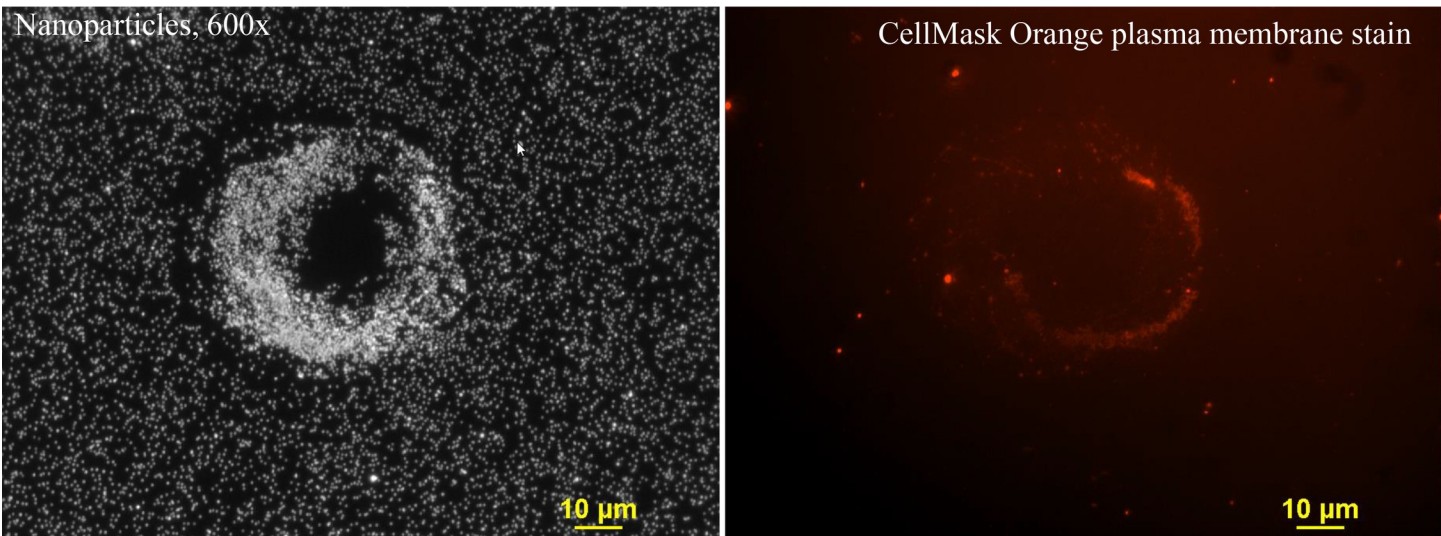

**Fig 4. A single ring structure was obtained with darkfield optics using a 60-x plan Fluor objective with an iris diaphragm (NA 0.55–0.90).** Panel A. A darkfield image of silver nanoparticles showing a concentrated ring structure. Panel B. A fluorescent image of the same field as Panel A after the slide was stained with Cell Mask Orange plasma membrane stain (red). A red ring of membrane proteins was observed corresponding to the circular formation of nanoparticles in Panel A. This suggests that cell membrane components remained on the slide after completion of mitosis. These residual materials were part of the cell membrane that preferentially attracted and/or bound silver nanoparticles that were deposited on the slide surface. The diameter of the circle was 53 μm, the inside circle was 27 μm, and the fluorescent ring was 37 μm. Magnification 600x., AgNP citrate 3ug/ml.

The cell size was difficult to evaluate accurately as many cells were larger than the field of view using a 60x objective. In addition, many cells may shrink during the fixation with paraformaldehyde. Cell size is also affected by culture conditions and cell concentration. It was estimated that the average size of the cells in their largest direction was about one third larger than the size of the nanoparticle circles.

The current studies used a combination of light microscopic techniques to visualize AgNP uptake, distribution within cells, transformation within cells, and the distribution AgNP deposited on the slides. The cells were transfected with organelle probes and then stained with fluorescence dyes to reveal the nuclei, cytoplasm, membranes, and organelles. (Figs 6 and 7)

The existence of silver nanoparticles rings is a new and interesting observation that relates to mitosis biology. Additional examples demonstrating the existence of these AgNP circles in a proliferating cellular culture were observed using darkfield and fluorescence microscopy and are shown in Figs 8–13. The four panels display of Fig 2 were shown in Figs 8–11 for clarity of the fluorescence staining and nanoparticle distributions.

## Discussion

Darkfield microscopy is considerably more sensitive than other light microscopic contrast techniques such as phase, differential interference (DIC), Nomarski and Hoffman DIC, making it suitable for observing nanoparticles less than 100 nm in size. When nanoparticles were observed with these other light microscopic contrast methods, they were imaging agglomerated particles with physical sizes above the microscope's diffraction limit of 220 nm. Darkfield microscopy was used previously to study nanoparticles in cells and on slides [13,15,17,21–25]. Although the current study used 80 nm AgNP, smaller sizes of AgNP can also be observed using darkfield microscopy. The darkfield microscopy optics used here consisted of a 60x oil objective with an iris diaphragm, a 20x multi-immersion oil lens (NA 0.75) and an oil

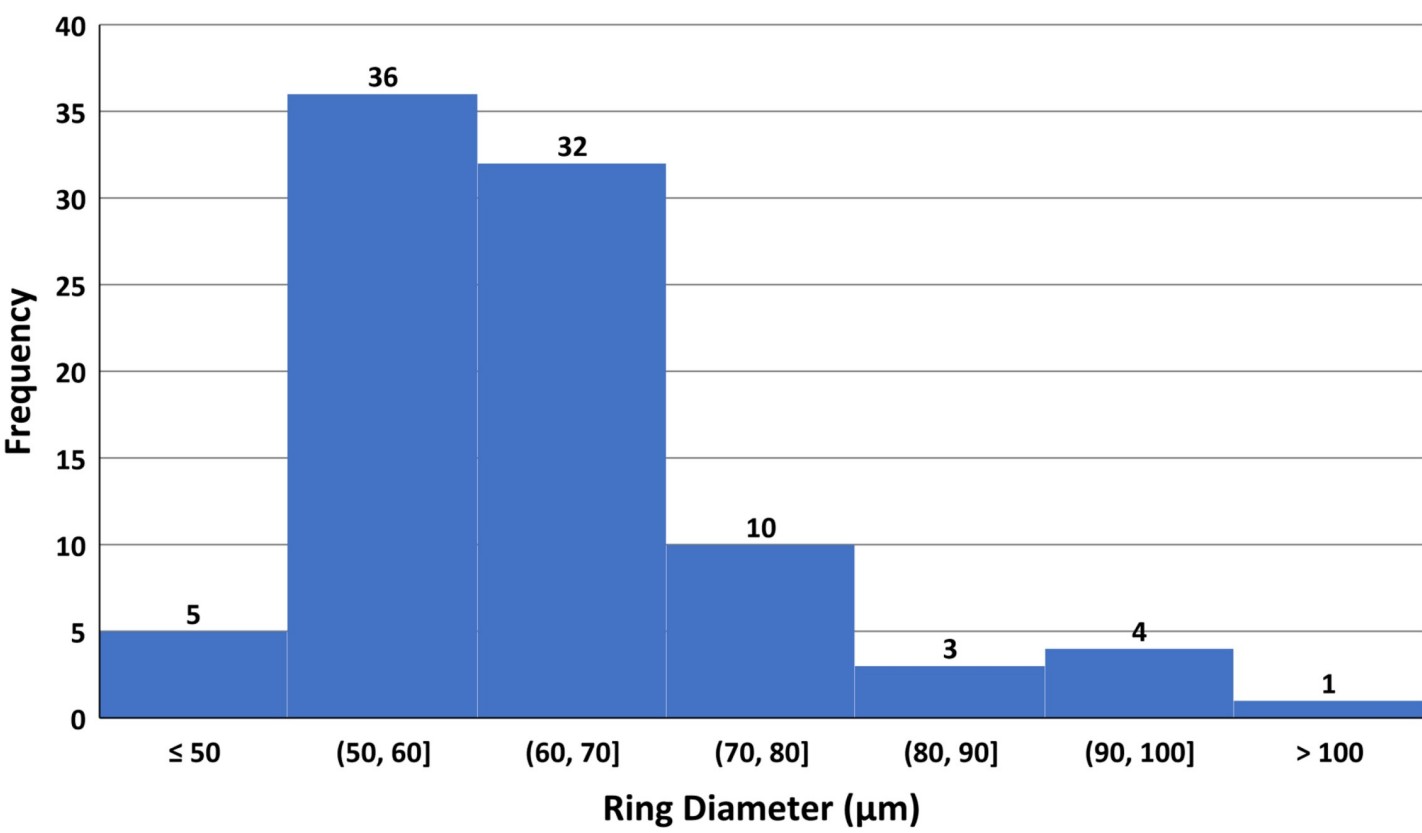

**Fig 5. Histogram consisting of 90 perpendicular diameter measurements from 45 nanoparticle circles.**

condenser. These components permitted the visualization of nanoparticles that would not have been observable with other microscopic contrast techniques.

### Intracellular distribution and agglomeration of nanoparticles

The distribution of AgNP inside the cytoplasm of ARPE-19 cells after in vitro treatment with AgNP has been described previously [13,15,17,25]. Initially, the AgNP traversed the cellular membrane into the cytoplasm, where the particles agglomerated around the nucleus or in the vicinity of the endoplasmic reticulum. Nanoparticles were not observed inside the nuclei, but they were observed surrounding the nuclei.

Using darkfield microscopy, particles located inside the cells were observed to reflect more light than particles located outside the cell. The brightness of intracellular AgNP reflects the collection of individual nanoparticles or formation of larger agglomerates as they were trafficked through the cell. Given that larger particles reflect more light, this indicates that the configuration of AgNP changed into larger agglomerates after they entered cells. In contrast, the AgNP outside the cell were usually observed to be spread uniformly on the culture slides in regions that did not contain cells.

It should be emphasized that dark field microscopy does not measure the actual physical size of the object or particle but the reflection of light from the space occupied by the particle. The net effect is the particle appears to be bigger than its actual physical size by using darkfield

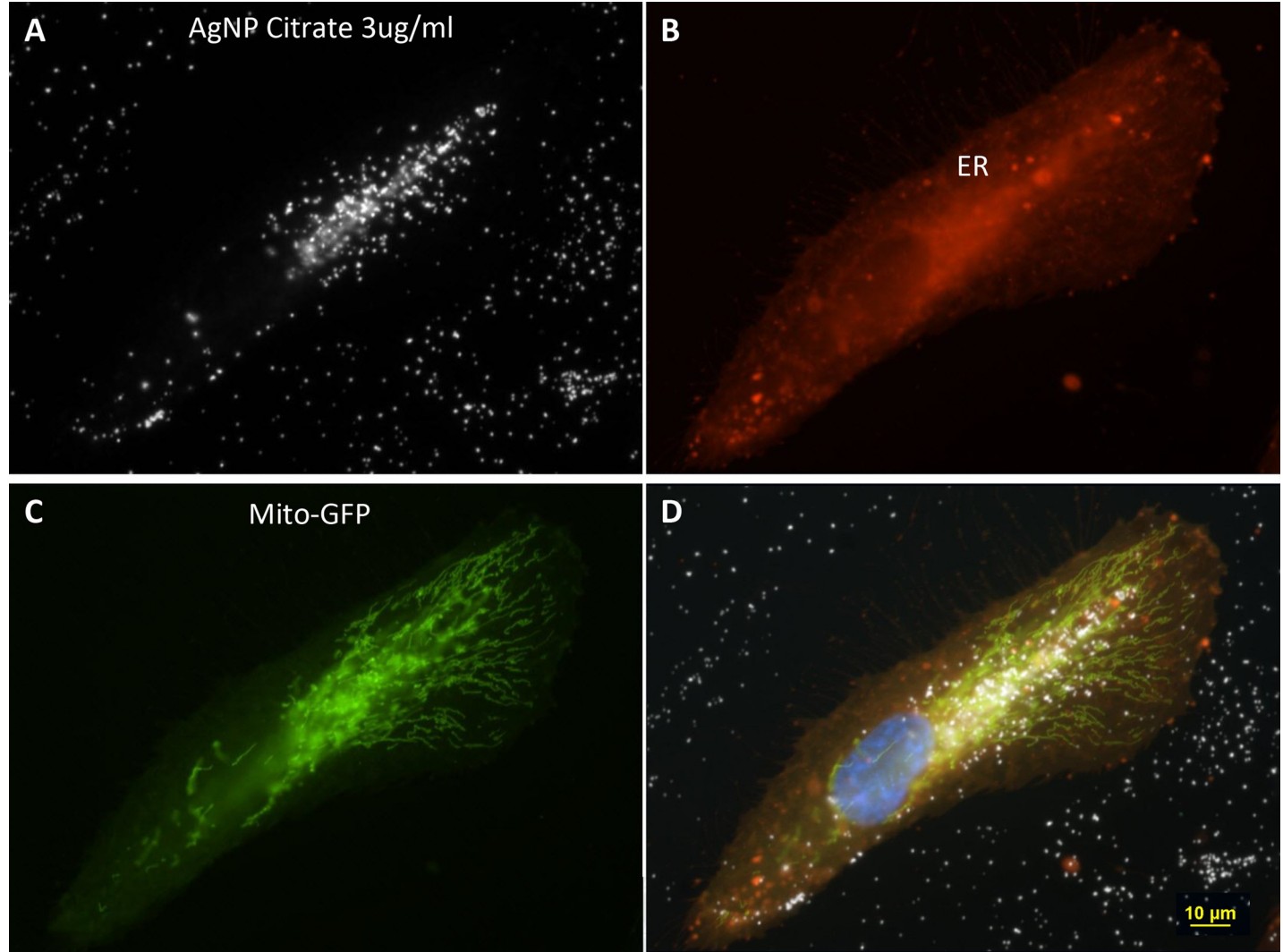

**Fig 6. Four different fields of 3ug/ml AgNP citrate obtained with darkfield optics using a 60-x plan Fluor objective with an iris diaphragm (NA 0.55–0.90).** Panel A. Intracellular nanoparticles are brighter than the extracellular particles. Panel B. The cytoplasm was stained with CellMask Orange plasma membrane stain (red). Panel C. Cells transfected with Mito-GFP (green). Panel D. A composite image of fields A, B, and C were combined with an image of DAPI stained nuclei (blue). Magnification 600x.

microscopy. The intracellular nanoparticles were not observed to change during the stages of the cell cycle or mitosis. The apparent size of extracellular nanoparticles by darkfield microscopy was about 0.37 μm, which is considerably larger than the 80 nm size of the source particles. The size of the clumped nanoparticles inside the cell appeared to be larger at 0.61 μm. The rings consisted of particles that appeared to have similar sizes to the extracellular particles dispersed on the slide surface. It was not possible to measure the individual size of particles in the rings due to the high particle density in the 3D ring structures.

## Formation of extracellular rings

Outside of the cells, there was a field of what appeared to be individual AgNP and an accumulation of concentrated AgNP in ring structures (Figs 1–4 and 8–13). We briefly reported these formations previously, but to our knowledge, these circular structures have not been further

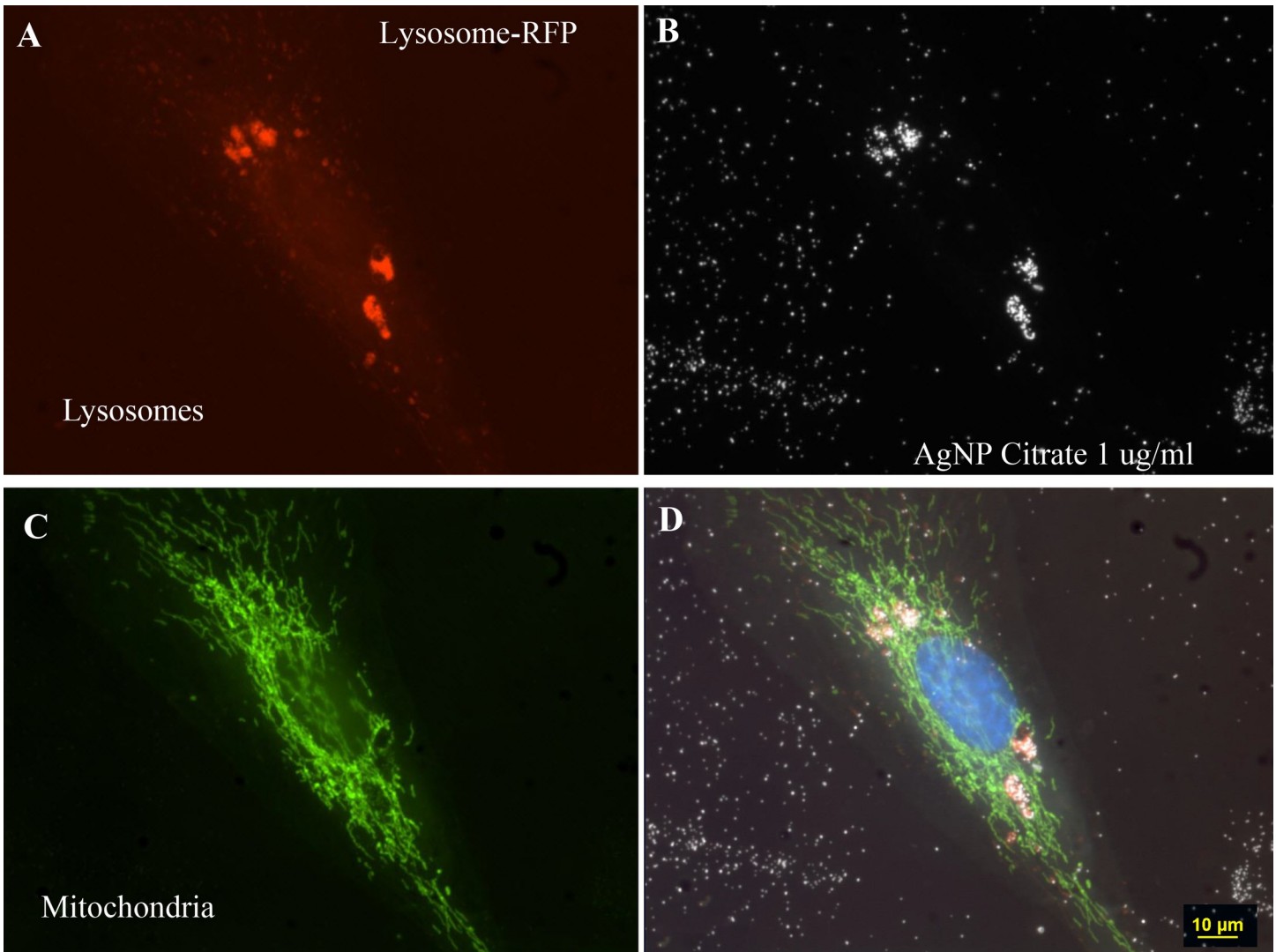

**Fig 7. Four different fields of 3ug/ml AgNP citrate obtained with darkfield optics using a 60-x plan Fluor objective with an iris diaphragm (NA 0.55–0.90).** Panel A. Cells were stained with Lysosome- RFP (red) to reveal the lysosomes. Panel B. Intracellular nanoparticles (white) were condensed into the identical regions occupied by lysosomes in A. Panel C. Cells transfected with Mito-GFP (green). Panel D. A composite image of fields A, B, and C were combined with DAPI stained nuclei (blue). Magnification 600x.

studied or evaluated in the scientific literature [25]. The circular structures consisted predominately of closely spaced nanoparticles which appeared to be the same size as individual extracellular particles on the slide. These extracellular cell-sized rings of AgNP were observed on the slides incubated with 3–10 µg/ml of nanoparticles (Figs 1–4 and 8–13). These circular structures could be observed for the following reasons: 1) the darkfield microscope provided resolution sufficient to observe nanoparticles below the 220 nm diffraction limit of a microscope and 2) the 3 to 10 µg/ml concentration of AgNP in suspension was high enough to provide an ample amount of these deposited nanoparticles.

The cell cycle is divided into stages of $G_1$, S, $G_2$ and M. Between $G_2$ and M a number of cellular events occur as the chromosomes condense and eventually line up on the metaphase plate. Mitosis is typically divided into 5 phases: prophase, prometaphase, metaphase, anaphase and telophase. Prophase is defined as the first stage in the transition between $G_2$ and mitosis.

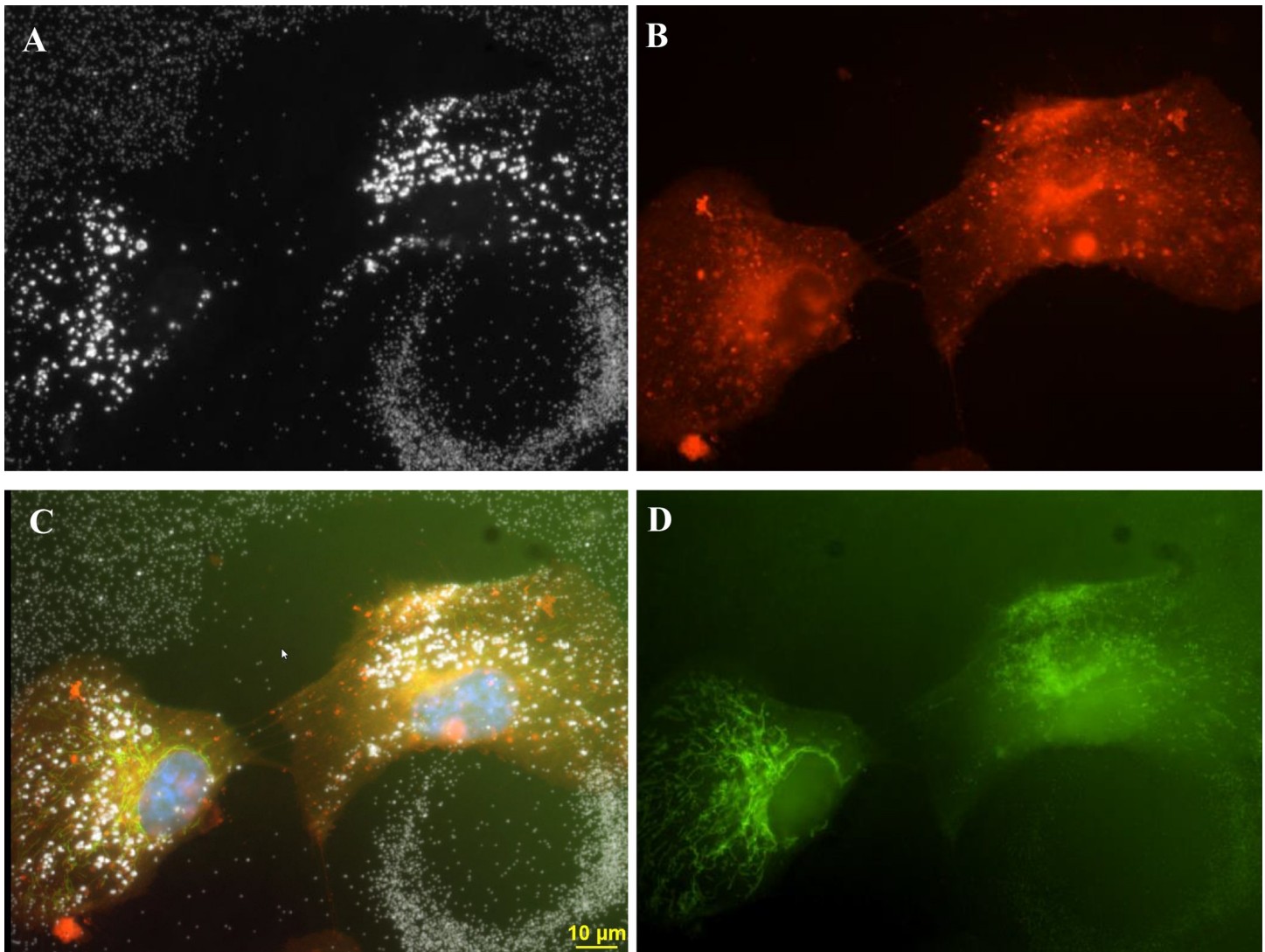

**Fig 8. The four-panel figure of cell shown in Fig 2A was treated with 3ug/ml AgNP.** The individual dark field and fluorescent images were combined into a composite image (C) Panel A. Darkfield image of a cell with an adjacent ring structure. Panel B. Cell stained with CellMask Orange plasma membrane stain. Panel C. Composite image of fields A, B, and C combined with DAPI stained nuclei (blue). Panel D. Cells transfected with Mito-GFP (green). Magnification 600x.

The sister chromosomes condense, and the mitotic spindle assembles between the two centromeres which replicate and move apart. The chromosomes attach to the spindle and begin movement to the metaphase equator. Cellular changes during prophase include condensed chromosomes, internalized cell surface markers, and disassembled intercellular membrane networks. The Golgi and endoplasmic reticulum fragment, and the cell begins to round up. The cells condense using focal adhesion proteins and actin filaments leaving some adhesion protein remnants behind on the slide after the completion of cell division [26–34].

It is thought that the positively charged substrate molecules promote cell adhesion by connecting the negatively charged cell membrane to the slide by electrostatic interaction [35]. Fig 4 shows the remnants of proteins in a circular pattern overlapping the AgNP ring that were stained with the membrane stain. CellMask Orange membrane plasma staining showed that the cytoplasmic material appeared in a circular pattern in the specific region occupied by the

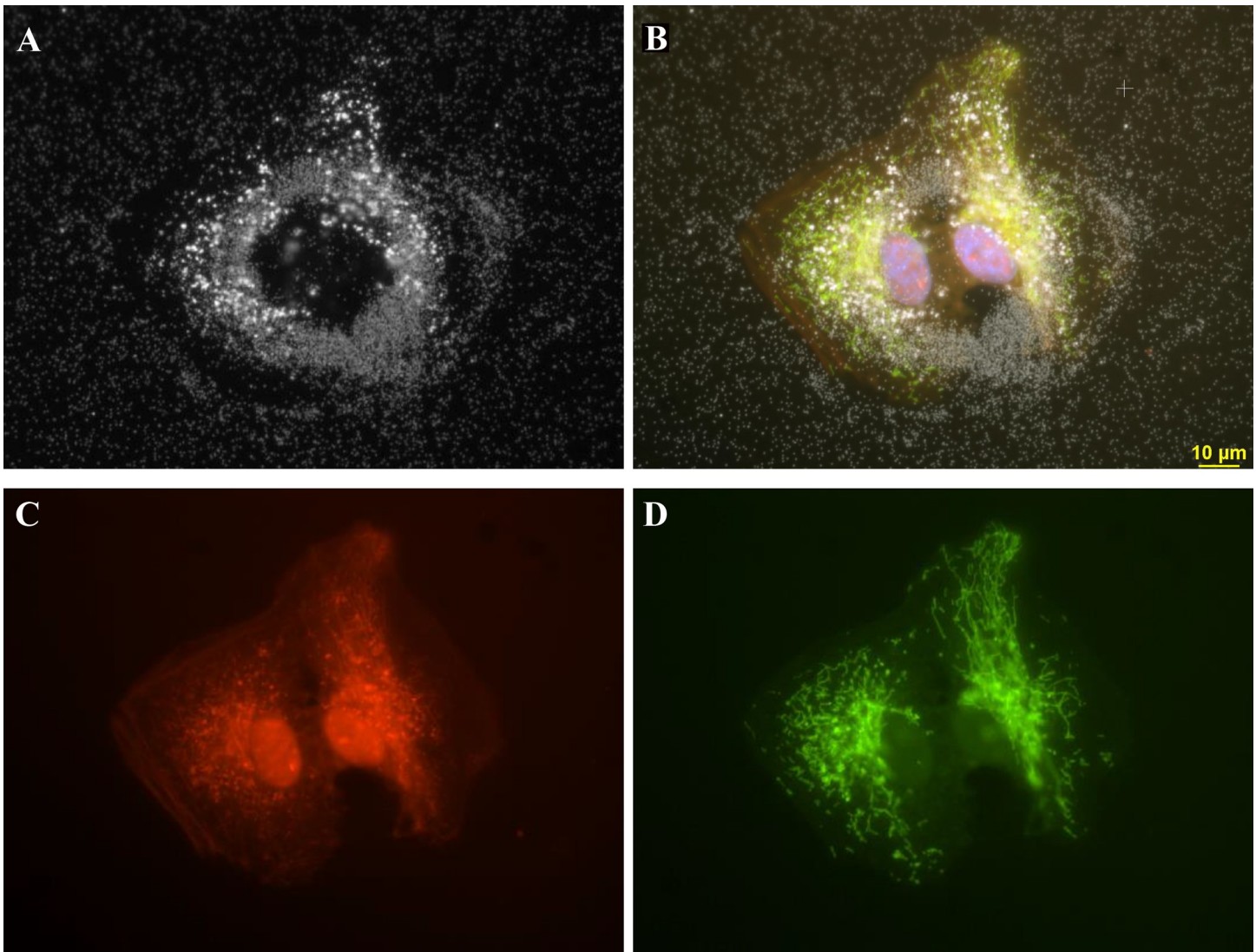

**Fig 9. The four-panel figure of cell shown in Fig 2B was treated with 3ug/ml AgNP.** Panel A. Darkfield image of a ring structure with 2 newly divided cells laying on top of ring structure. Particles in cell are larger and brighter than particles in ring or extracellular. Panel B. The individual dark field (A) and fluorescent images (C, D) were combined into a composite image (B). DAPI stained nuclei (blue) image was added to the composite. Panel C. Cell stained with CellMask Orange plasma membrane stain. Panel D. Cells transfected with Mito-GFP (green). Magnification 600x.

AgNP ring. These circular structures may have been formed by the interaction of nanoparticles with adhesion proteins released during mitotic contraction process [26–34]. These residual adhesion proteins may accumulate negatively charged nanoparticles to form these circular structures.

The relationship between the ring structures and mitosis could be a useful parameter to describe the events occurring between prophase and metaphase. This observation may be correlated with the other cellular events involving microtubules and chromosome condensation. During prophase, and prior to the chromosomes lining up on the metaphase plate, the cells round up, halt their movement toward the center, and a central hole is formed where the chromosomes line up. A thick ring of adherent proteins at the cell's periphery becomes active to move the ends of the cell into the center, which perhaps leaves a residue that preferentially

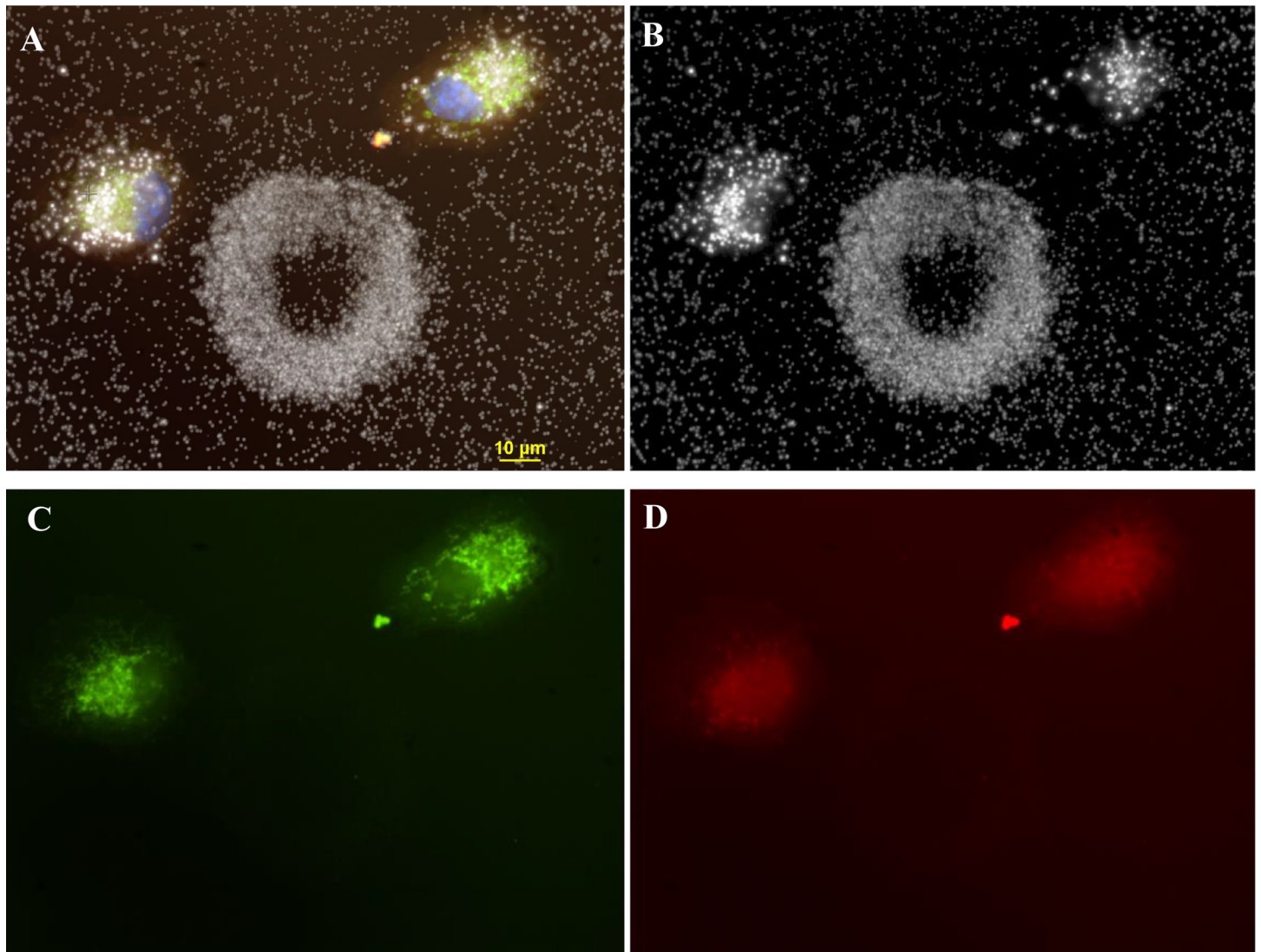

**Fig 10. The four-panel figure of cell shown in Fig 2C was treated with 3ug/ml AgNP.** Panel A. The individual dark field (B) and fluorescent images (C, D) were combined into a composite image (A). DAPI stained nuclei (blue) image was added to the composite image. Panel B. Darkfield image of a ring structure with 2 newly divided cells adjacent to the top of ring structure. Panel C. Cells transfected with Mito-GFP (green). Panel D. Cell stained with CellMask Orange plasma membrane stain. Magnification 600x.

attracts silver nanoparticles and creates the nanoparticle configurations observed here. The extent that the rings are oval shaped may be because the mitotic cells are not totally circular but appear to differ between the vertical and horizontal axis with the lining up of chromosomes on the metaphase plate. The size and thickness of the ring walls may be indicative of how mitosis proceeded and have significance for normal or abnormal mitosis.

The exact composition of the proteins or cellular materials that was stained by CellMask Orange membrane plasma stain is not known. The types of adhesion proteins that were left on the slide and bound nanoparticles is an area for further study. It would be interesting to correlate the circular structures with stages of prophase and prometaphase, and with the biochemical events that occur during mitosis.

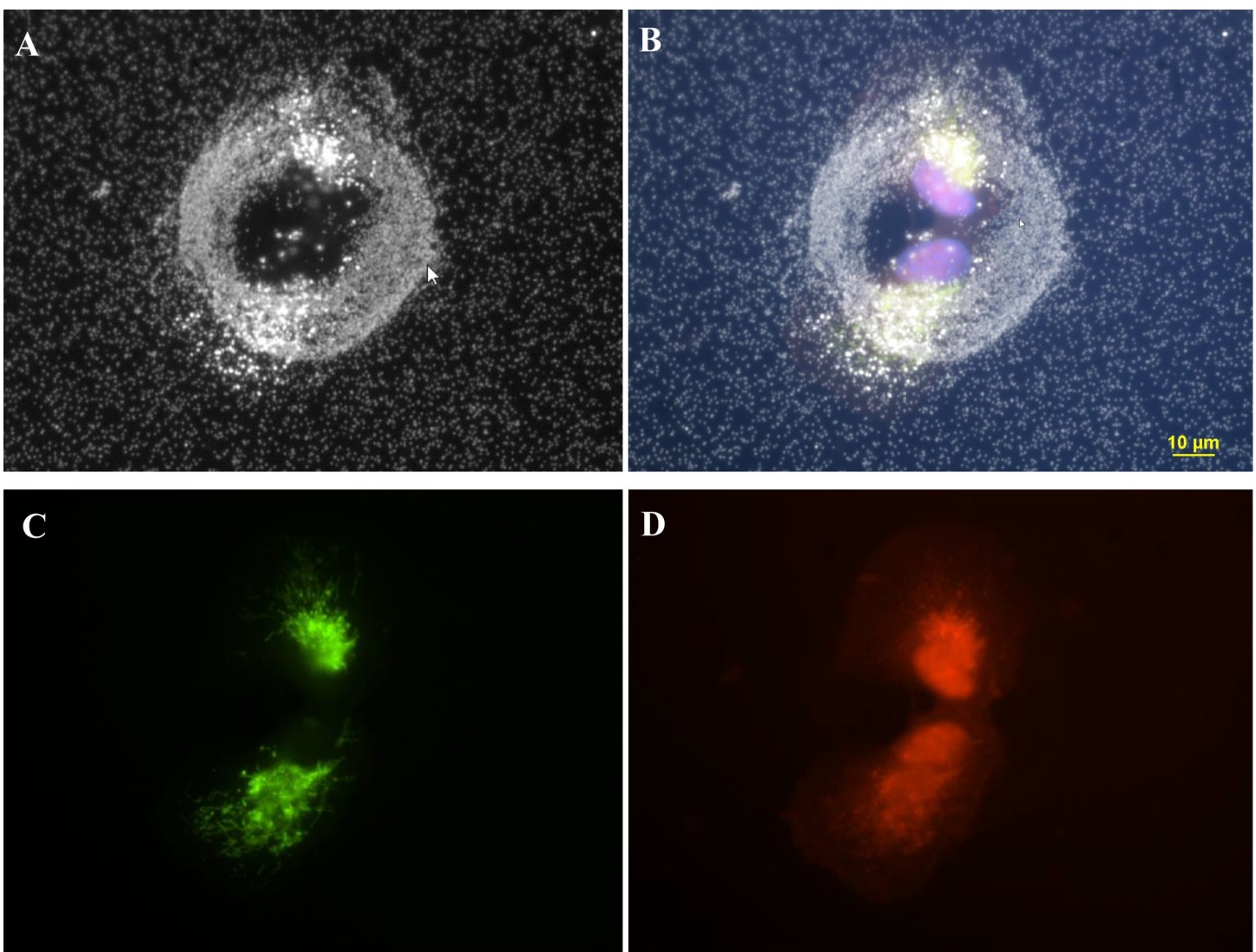

**Fig 11. The four-panel figure of cell shown in Fig 2D was treated with 3ug/ml AgNP.** Panel A. Darkfield image of a ring structure with 2 newly divided cells on top of ring structure. Panel B. Composite of individual dark field (A) and fluorescent images (C, D) were combined into a composite image (B). DAPI stained nuclei (blue) was added to the composite. Panel C. Cells transfected with Mito-GFP (green). Panel D. Cell stained with CellMask Orange plasma membrane stain. Magnification 600x.

### Measurements of extracellular circular structures

Typically, the cultured ARPE-19 epithelial cells are thin, elongated cells with little vertical thickness. These cells average around 85 microns in length (1/3 larger than the circles) but some cells can be over 150 microns long in one direction. During cell division, both intracellular actin filaments and external cellular adhesion proteins are used to move the cell across the slide surface during mitotic cellular contraction. In this process, the cell changes its footprint from a thin, relatively flat large structure to a smaller, spherical ball-like structure [36,37]. This change results in a loss of observable structural features inside the cell when viewing the cells in a 2D image plane. The size of telophase mitotic cells was found to be approximately 35 μm, which is about one third the size of a log growing cell. The measurement of 45 extracellular nanoparticle circles was found to be 62.5 +/- 12 μm.

Due to the thickness of the mitotic cells, a cellular image was obtained that was not completely in focus. The image resolution of the two rounded mitotic cells observed in Fig 2

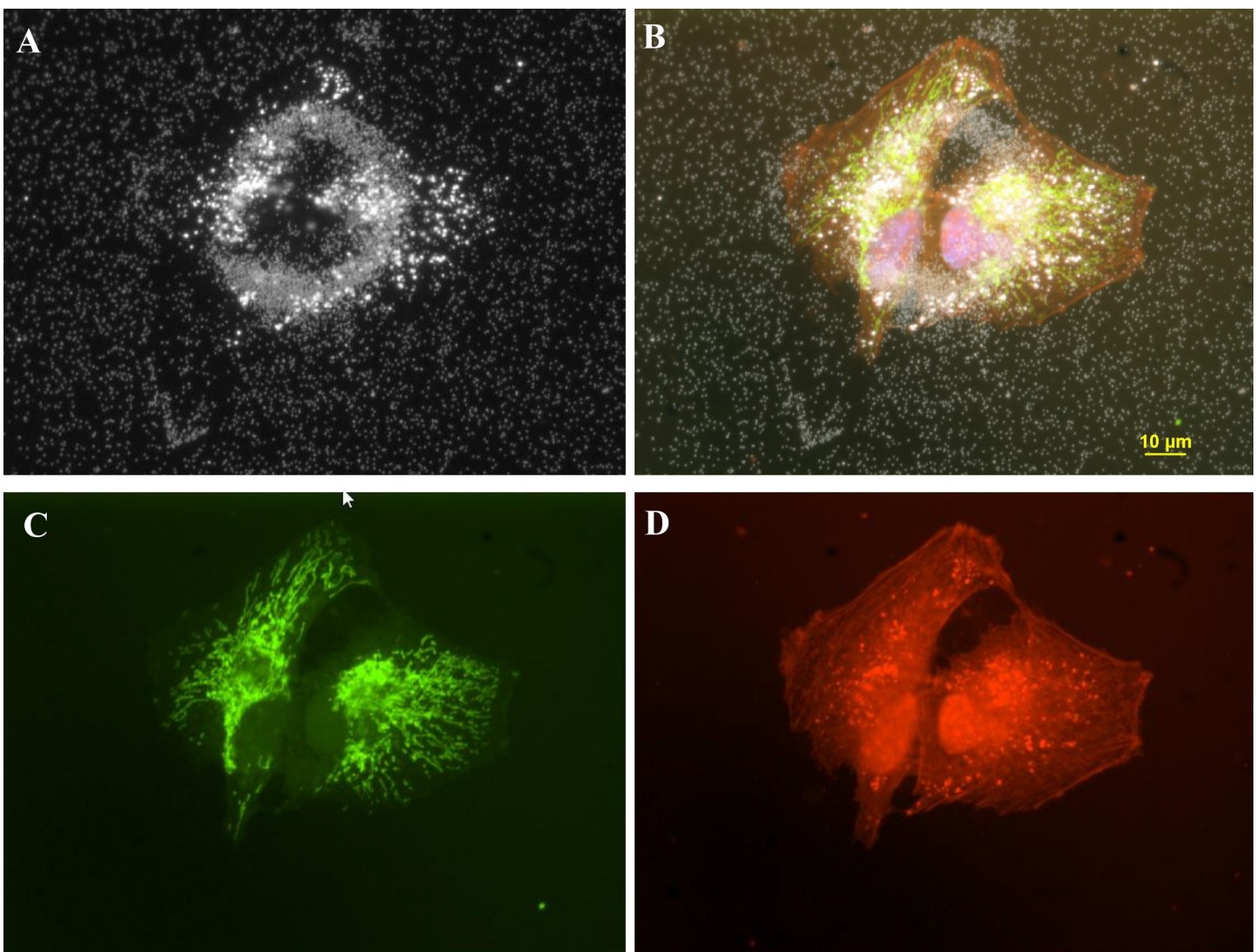

**Fig 12. The four-panel figure of cell treated with 3ug/ml AgNP.** Panel A. Darkfield image of a ring structure with 2 newly divided daughter cells on top of ring structure. Panel B. The individual dark field (A) and fluorescent images (C, D) were combined into a composite image (B). DAPI stained nuclei (blue) image was added to the composite. Panel C. Cells transfected with Mito-GFP (green). Panel D. Cell stained with CellMask Orange plasma membrane stain. Magnification 600x.

was not as sharp as flatter non-mitotic cells shown in Figs 3, 6 and 7. After cell division, the newly formed daughter cells settled onto the slide to repeat this growth and division process. The images of the AgNP circular structures shown in Figs 1–4 may have been derived from recently divided cells that left behind adhesion proteins that bound nanoparticles in a circular pattern. Generally, there were 0, 1, or 2 cells associated with these large nanoparticle ring structures (Figs 2 and 8–10). These small pairs of cells in Figs 2 and 9–12 were almost identical in morphology suggesting that they were newly divided mitotic cells. It appears that most of the new "daughter" cells do not move far away from the large circular structure. The cells sometimes settled into the region occupied by the circle or nearby the circle (Fig 2B–2D).

The AgNP (citrate) used in the current study were negatively charged particles, which may help in their binding to positively charged proteins. Studies with 3T3 cells and CHO cells show positively charged adhesion proteins [38,39]. These focal adhesion proteins appear to have a

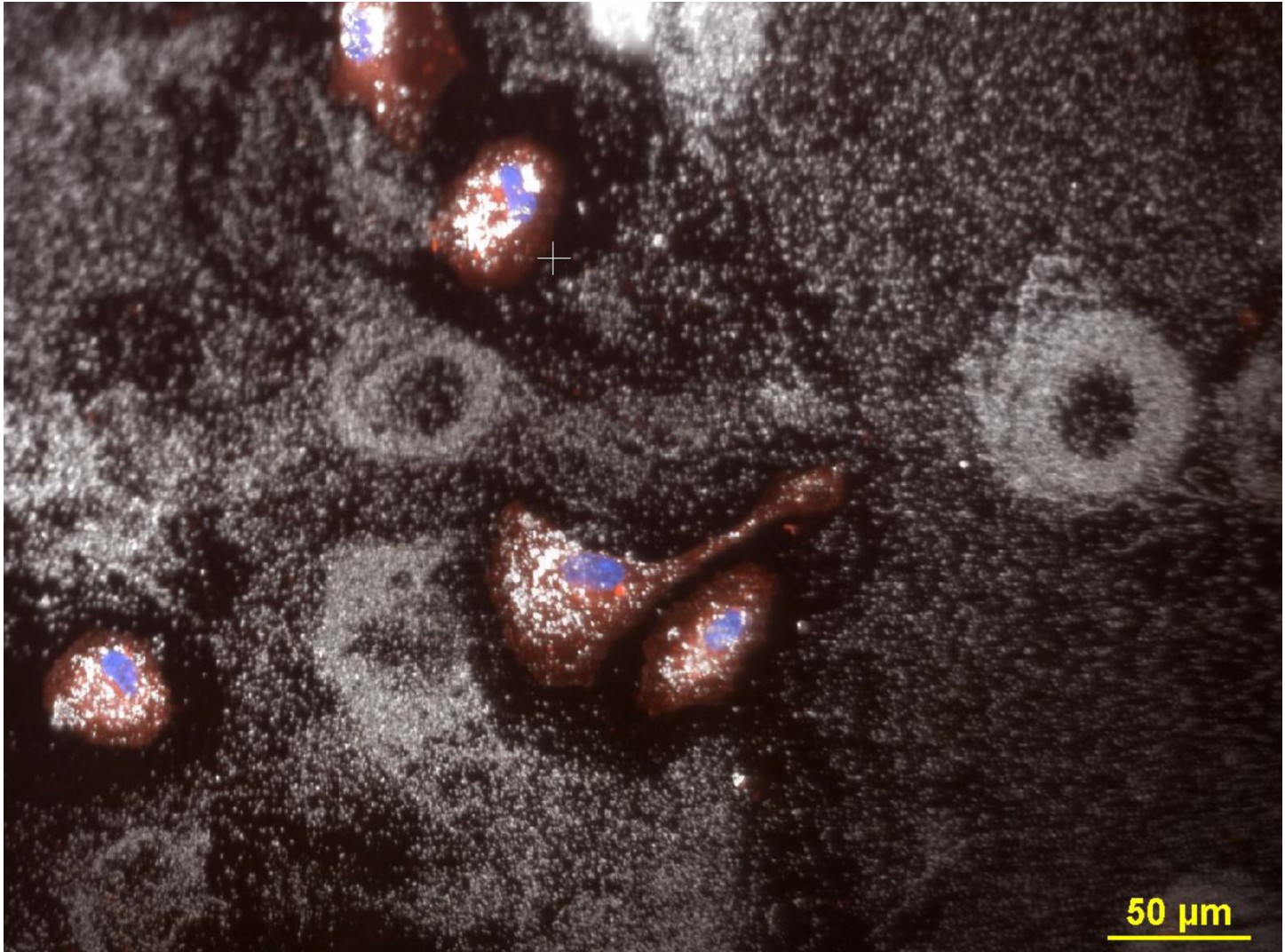

**Fig 13. ARPE-19 cells incubated with 10 µg/ml AgNP citrate.** The cells were transfected with mitochondria-GFP (green), stained with CellMask Orange plasma membrane stain (C10045, red), fixed with 4% PF, and mounted with Prolong gold that contained DAPI to stain the nuclei (blue). Field shows a dense accumulation of nanoparticles, two ring structures and 5 cells. The nanoparticles inside the cell are brighter than the particles outside the cell. The image consists of a combination of one darkfield and three fluorescence images that were acquired sequentially and then combined into one composite. Magnification 200x.

maximum concentration just prior to the cells entering the $G_2$ phase of the cell cycle. The focal adhesions then appear to degrade as the cell progresses from late S phase to $G_2$ phase. By the end of $G_2$ phase, the focal adhesions had disassembled, possibly leaving some adhesion proteins at the periphery [40]. The charge on these proteins, or other unique biophysical characteristics, may attract the negatively charged nanoparticles. Fig 4 shows remnants of proteins that were stained with the CellMask Orange membrane plasma stain.

## Concentration of nanoparticles applied to cells

As described by Stokes' law, and due to AgNP agglomeration in suspension, the number of particles depositing on the surface of the slide increased over time. Unlike chemicals in solution, the effective concentration that the cell was exposed to could be much higher than what was added to the medium due to gravity sedimentation. The observation of particle deposition

across the slide's surface in addition to the appearance of large round extracellular structures consisting of nanoparticles is suggestive of a high particle concentration.

The experiments described in this manuscript used suspensions of AgNP that were obtained at a concentration of 1 mg/ml. This concentration is equivalent to around $10^{12}$ particles per ml. A one-thousand-fold dilution to achieve 1 µg/ml dosing suspension results in about 100 million particles ($10^8$), while 10 µg/ml results in a solution which contains 1 billion particles per ml ($10^9$). The darkfield image data revealed that there was an excess of particles on the slide at concentrations of $10^8$ and $10^9$ particles/ml. Higher concentrations of particles in the range of 30 µg/ml coated the cell surface with a film of particles, which may result in a physical barrier that could result in a suffocation of the cell transport and inhibition of various biochemical processes. A similar observation of cellular inhibition was reported using titanium dioxide nanoparticles that induced cytotoxicity by inhibiting ion exchange [41].

The high concentration of nanoparticles combined with darkfield microscopy allowed us to observe structures like forensic tests that look for fingerprints by dusting a sample. However, the use of these concentrations below 3 µg/ml did not show these types of circles on the slides. A concentration of 1 µg/ml dose of AgNP applied to cells was equivalent to adding $10^8$ particles/ml and that concentration was sufficient to visualize particles in cells or on the slides. From these observations, it appears that the amount of AgNP used in these experiments may have been excessive, allowing us to visualize these extracellular circles. It should be noted that the concentrations used in these experiments were about 1/10 to 1/100[th] of the concentrations that other investigators have used in their published studies. For example, Carlson et al (2008) studied AgNP cytotoxicity over a dose range of 10–75 µg/ml, and Mukherjee et al (2012) over a range of 25–100 µg/ml [42,43] in their experiments.

## Summary

This study evaluated the presence of extracellular circular AgNP formations in vitro. AgNP were observed on the slide surfaces or within cells using darkfield microscopy. The individual AgNP were usually spread uniformly on the culture slides in areas that did not contain cells. Occasionally, circular structures consisting of AgNP with the approximate size of adherent cells were observed on the slide of cells treated with either 3 or 10 ug/mL AgNP suspensions. It is hypothesized that these ring structures originated during the mitotic cytokinesis contraction process by which negatively charged AgNP combined with adhesion proteins that were released during mitotic contraction. This observation may be useful for future studies in cell mitosis, cell biology and nanoparticle research.

## Acknowledgments

Special thanks are extended to John Rogers, Chris Lau, Dave Herr and Alice Goldstein-Plesser for their helpful comments and encouragement for many aspects of the paper. We wish to thank Phil Hartig and Mary Cardon for help with the transfection experiments.

## Author Contributions

**Conceptualization:** Robert M. Zucker, William K. Boyes.

**Data curation:** Robert M. Zucker, Laura L. Degn, William K. Boyes.

**Formal analysis:** Robert M. Zucker.

**Funding acquisition:** Robert M. Zucker.

**Investigation:** Robert M. Zucker, Jayna Ortenzio, Laura L. Degn, William K. Boyes.

**Methodology:** Robert M. Zucker, Jayna Ortenzio, Laura L. Degn, William K. Boyes.

**Project administration:** Robert M. Zucker, Laura L. Degn.

**Resources:** Robert M. Zucker, William K. Boyes.

**Software:** Robert M. Zucker.

**Supervision:** Robert M. Zucker, Laura L. Degn.

**Validation:** Robert M. Zucker, William K. Boyes.

**Visualization:** Robert M. Zucker.

**Writing – original draft:** Robert M. Zucker, William K. Boyes.

**Writing – review & editing:** Robert M. Zucker, Jayna Ortenzio, Laura L. Degn, William K. Boyes.

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
