## [Decision Letter · Decision Letter 0]

20 Jul 2020

PONE-D-20-19210

Detection of Large Extracellular Silver Nanoparticle Rings observed during Mitosis using Darkfield Microscopy

PLOS ONE

Dear Dr. Zucker,

Thank you for submitting your manuscript to PLOS ONE. After careful consideration, we feel that it has merit but does not fully meet PLOS ONE’s publication criteria as it currently stands. Therefore, we invite you to submit a revised version of the manuscript that addresses the points raised during the review process.

We look forward to receiving your revised manuscript.

Kind regards,

Amitava Mukherjee, ME, Ph.D.

Academic Editor

PLOS ONE

Journal Requirements:

Reviewers' comments:

Reviewer's Responses to Questions

**Comments to the Author**

1. Is the manuscript technically sound, and do the data support the conclusions?

Reviewer #1: Partly

2. Has the statistical analysis been performed appropriately and rigorously? 

Reviewer #1: No

3. Have the authors made all data underlying the findings in their manuscript fully available?

Reviewer #1: Yes

4. Is the manuscript presented in an intelligible fashion and written in standard English?

Reviewer #1: Yes

5. Review Comments to the Author

Reviewer #1: PLOS ONE

Detection of Large Extracellular Silver Nanoparticle Rings observed during Mitosis

using Darkfield Microscopy

Manuscript Number: PONE-D-20-19210

Article Type: Research Article

In this article describe a technique that allows particles to be observed that would normally be invisible using other microscopic contrast techniques. The authors incorporated earlier suggestions by reviewers. There still seems to be scope of further improvement. The authors can go through the following points and revise the article.

1. Going through the abstract, the article and the conclusion, the authors have emphasised on the unusual appearance of silver nanoparticles around the cells undergoing mitosis. The authors leave the readers lost mentioning that these could be useful for future investigations. Perhaps the authors could emphasise more on this utility they foresee, thus, contributing to the scientific community. A histogram would definitely be a desired add-on.

2. The size of the nanoparticles in relation to the stage of the mitosis- can they be tabulated to give a rough idea to gauge the stage of cell division from the particle size, approximately?

3. Perhaps, some more data to establish the technique could be provided.

4. Also, the authors could go through the manuscript once again and remove additional texts.

6. PLOS authors have the option to publish the peer review history of their article (what does this mean?). If published, this will include your full peer review and any attached files.

Reviewer #1: No

---

## [Author Response · Author response to Decision Letter 0]

21 Aug 2020

Reviewer #1: PLOS ONE

Detection of Large Extracellular Silver Nanoparticle Rings observed during Mitosis

using Darkfield Microscopy

Manuscript Number: PONE-D-20-19210

Article Type: Research Article

Dear Dr Mukherjee , 

Thank you for giving us an opportunity to revise our manuscript, to answer ther reveiwers concens and make it more comprehnsible for the PLOS1 audinece. We have addressed your concerns and hope it is now suitable for publication in PLOS 1. 

Comment: There still seems to be scope of further improvement. The authors can go through the following points and revise the article.

1. Going through the abstract, the article and the conclusion, the authors have emphasized on the unusual appearance of silver nanoparticles around the cells undergoing mitosis. The authors leave the readers lost mentioning that these could be useful for future investigations. Perhaps the authors could emphasize more on this utility they foresee, thus, contributing to the scientific community. A histogram would definitely be a desired add-on.

Response: An explanation for how the nanoparticles rings originate during prophase is presented in the discussion and mentioned in the abstract. A new histogram (Figure 5) of ring sizes has been added as suggested by the reviewer to summarize the sizes of rings observed. An interpretation of the formation and thickness of the rings occurring during the prophase stage of mitosis has been proposed in the discussion. Additional data on the size of the rings and their significance have been provided in the discussion. The ring structure may be integrated with events that occur during prophase as the cell transitions between G2 restriction point and metaphase. 

The relationship of these ring structures appears to be another useful parameter to describe the events between prophase and metaphase which may correlate to the other events involving microtubules and chromosome condensation. 

2. The size of the nanoparticles in relation to the stage of the mitosis- can they be tabulated to give a rough idea to gauge the stage of cell division from the particle size, approximately

Response. The nanoparticles were not observed to be different during the stage of the cell cycle or mitosis. Particles inside the cell are brighter and thus considered larger and clumped compared to particles outside the cell or in the rings. The measurement of nanoparticles by darkfield microscopy was found to be about 0.37 μm. The apparent size of the nanoparticles inside the cell are clumped and appear to be larger at 0.61 μm. These data are incorporated into the text of the manuscript. It should be emphasized that dark field microscopy does not measure the actual size of the object or particle but shows the reflection of light from the location of the particle. The darkfield observation of Ag nanoparticles show the particles to appear bigger than their actual physical size. 

The rings consist of particles that appear to have similar sizes to the extracellular particles distributed on the slide surface. It was not possible to measure the individual size of particles in the rings due to their high density, clumping and their location in a 3d structure. 

3. Perhaps, some more data to establish the technique could be provided. 

Response: A histogram of the rings structures was provided in in a new Figure 5. We also measured the size of the nanoparticles using Nikon elements by darkfield imaging. It was found that these particles have an apparent size of 0.37 μm by dark field microscopy, through measuring the area of reflected light as opposed to actual physical size of the particle which was described as 80 nm by the manufacturer. The darkfield methodology can show particles below the diffraction limit of the microscope. The resulting image creates a spot that is larger than its physical size. 

4. Also, the authors could go through the manuscript once again and remove additional texts.

Response –The text has been revised and additional text removed.

---

## [Decision Letter · Decision Letter 1]

8 Sep 2020

PONE-D-20-19210R1

Detection of Large Extracellular Silver Nanoparticle Rings observed during Mitosis using Darkfield Microscopy

PLOS ONE

Dear Dr. Zucker,

Thank you for submitting your manuscript to PLOS ONE. After careful consideration, we feel that it has merit but does not fully meet PLOS ONE’s publication criteria as it currently stands. Therefore, we invite you to submit a revised version of the manuscript that addresses the points raised during the review process.

We look forward to receiving your revised manuscript.

Kind regards,

Amitava Mukherjee, ME, Ph.D.

Academic Editor

PLOS ONE

Reviewers' comments:

Reviewer's Responses to Questions

**Comments to the Author**

1. If the authors have adequately addressed your comments raised in a previous round of review and you feel that this manuscript is now acceptable for publication, you may indicate that here to bypass the “Comments to the Author” section, enter your conflict of interest statement in the “Confidential to Editor” section, and submit your "Accept" recommendation.

Reviewer #1: All comments have been addressed

2. Is the manuscript technically sound, and do the data support the conclusions?

Reviewer #1: Partly

3. Has the statistical analysis been performed appropriately and rigorously? 

Reviewer #1: Yes

4. Have the authors made all data underlying the findings in their manuscript fully available?

Reviewer #1: Yes

5. Is the manuscript presented in an intelligible fashion and written in standard English?

Reviewer #1: No

6. Review Comments to the Author

Reviewer #1: The authors have answered all the queries as asked.

However, the way of writing of the article needs to be checked further. Its not lucid and hence, reading the article doesnot garner much interest.It would be nice if the authors could go through again and revise the language and the style of writing as per the journal standards.

7. PLOS authors have the option to publish the peer review history of their article (what does this mean?). If published, this will include your full peer review and any attached files.

Reviewer #1: No

---

## [Author Response · Author response to Decision Letter 1]

19 Sep 2020

Re: PONE-D-20-19210R1

Amitava Mukherjee, ME, Ph.D.

Academic Editor

PLOS ONE

Dear Dr Mukherjee,

Thank you and your peer reviewers for the careful evaluation of our manuscript entitled: “Detection of Large Extracellular Silver Nanoparticle Rings observed during Mitosis using Darkfield Microscopy”. For our previous revision, the reviewers indicated that the revised manuscript had addressed all the technical issues identified, but that the writing style and editorial quality were insufficient. The reviewers, however, did not identify any specific sections of the manuscript that needed attention.

Today, we are submitting a revised version of the manuscript which has been thoroughly revised to improve the editorial quality. Among other changes, we have removed redundancies, simplified and corrected sentence structures, corrected grammar, improved the consistency of acronyms and symbols, and corrected reference styles. We submit a “marked” version in which changes to the manuscript are indicated using the MS Word “track change” feature. You will see that the editorial revisions were extensive. We also submit a “clean” version in which all the changes were accepted. Overall, we believe, the quality and readability of the manuscript were improved, which should make it more interesting for the readers of PLOS ONE. 

We hope that your reviewers will observe the improved quality of the manuscript and determine that it is now acceptable for publication in PLOS ONE.

Sincerely,

Robert Zucker, PhD

U.S. Environmental Protection Agency

Research Triangle Park, NC 27701

Email: zucker.robert@epa.gov

---

## [Editor Report · Decision Letter 2]

23 Sep 2020

Detection of Large Extracellular Silver Nanoparticle Rings observed during Mitosis using Darkfield Microscopy

PONE-D-20-19210R2

Dear Dr. Zucker,

We’re pleased to inform you that your manuscript has been judged scientifically suitable for publication and will be formally accepted for publication once it meets all outstanding technical requirements.

Kind regards,

Amitava Mukherjee, ME, Ph.D.

Academic Editor

PLOS ONE
---

## [Editor Report · Acceptance letter]

8 Oct 2020

PONE-D-20-19210R2 

Detection of large extracellular silver nanoparticle rings observed during mitosis using darkfield microscopy 

Dear Dr. Zucker:

I'm pleased to inform you that your manuscript has been deemed suitable for publication in PLOS ONE. Congratulations! Your manuscript is now with our production department. 

Kind regards, 

on behalf of

Professor Dr. Amitava Mukherjee 

Academic Editor

PLOS ONE